# A Study on the Control Method of 6-DOF Magnetic Levitation System Using Non-Contact Position Sensors

**DOI:** 10.3390/s23020905

**Published:** 2023-01-12

**Authors:** Dong-Hoon Jung, Jong Suk Lim

**Affiliations:** 1School of Mechanical, Automotive and Robot Engineering, Halla University, Wonju 26404, Republic of Korea; 2Vehicle Electrification Research Center, Korea Automotive Technology Institute, Daegu 43011, Republic of Korea

**Keywords:** edge bread remove (EBR) process, integrated driving control, non-contact position sensor, 6-DOF magnetic levitation system

## Abstract

Recently, due to the development of semiconductor technology, high-performance memory and digital convergence technology that integrates and implements various functions into one semiconductor chip has been regarded as the next-generation core technology. In the semiconductor manufacturing process, various motors are being applied for automated processes and high product reliability. However, dust and shaft loss due to mechanical friction of a general motor system composed of motor-bearing are problematic for semiconductor wafer processing. In addition, in the edge bread remove (EBR) process after the photoresist application process, a nozzle position control system for removing unnecessary portions of the wafer edge is absolutely necessary. Therefore, in this paper, in order to solve the problems occurring in the semiconductor process, a six-degrees-of-freedom (6-DOF) magnetic levitation system without shaft and bearing was designed for application to the semiconductor process system; and an integrated driving control algorithm for 6-DOF control (levitation, rotation, tilt (Roll–Pitch), X–Y axis movement) using the force of each current component derived through current vector control was proposed. Finally, the 6-DOF magnetic levitation system with the non-contact position sensors was fabricated and the validity of the 6-DOF magnetic levitation control method proposed in this paper was verified through a performance test using a prototype.

## 1. Introduction

Semiconductors are next-generation core parts that are being applied to various industries such as home appliances, automobiles, robots, and IT, and the size of the market is continuously growing. Accordingly, a number of motors having advantages of high output, high efficiency, low vibration, and miniaturization are applied to the semiconductor manufacturing process for process automation and product quality improvement [1]. In general, motor systems have problems such as mechanical friction of bearings and shaft wear and dust caused by them. In particular, dust generated from the shaft and bearings of the Spin coater, a motor system used in the photoresist coating process during the semiconductor wafer process, causes a fatal problem. To solve this problem, a dust absorption structure is additionally installed and operated in the semiconductor wafer processing system. In addition, in the EBR process system for removing unnecessary portions of the wafer edge after the photoresist coating process, a nozzle position control system is additionally required. This paper is a study of the design and control method of a 6-DOF magnetic levitation system using a non-contact position sensors that can replace the spin coater used in the existing semiconductor wafer process. The six-degrees-of-freedom magnetic levitation system that combines magnetic levitation and rotational motion was designed by using the force generated by independent current components obtained through vector control of a permanent magnet motor. Then, the control methods of levitation, tilt, rotation, and movement for 6-DOF motion were studied, and the system-integrated driving control algorithm using the combination of forces that can be generated in the system based on the control method for each degree of freedom was proposed. Finally, the validity of the integrated driving control method of the 6-DOF system proposed in this paper was verified through a performance test using a prototype of the 6-DOF magnetic levitation system.

## 2. Design of 6-DOF Magnetic Levitation System

The motor of the 6-DOF magnetic levitation system proposed in this paper has the same structure as the axial flux permanent magnet motor as shown in Figure 1a, but it is a system in which the shaft and bearing are removed by using magnetic levitation to replace the existing spin coater. In general, current vector control used to control a permanent magnet motor converts the current applied to the stator into a d–q-axis current, and each current can be controlled independently. Therefore, the q-axis current can be used for rotational motion of the 6-DOF magnetic levitation system, and the d-axis current can be used for magnetic levitation through the repulsive force between the permanent magnets of the rotor [2,3,4,5,6,7]. That is, the rotation and levitation control of the system is possible through the vector control of the current applied to the stator [8,9]; and the stator is divided into 4 parts for additional axial movement, such as tilt (roll–pitch) and movement in the x–y axis. It is designed as a 4C1M (4 Controller 1 Motor) system consisting of 4 controllers in one motor to enable different vector control for the 3-phase current for each part of the stator [10,11]. Additionally, the sensor base, composed of gap sensors for measuring the levitation position of each part of the stator and non-contact optical encoders for measuring the rotational position of the rotor was designed in the center of the system for control of each axis movement. Figure 1b shows a cross-section of the 6-DOF magnetic levitation system and the structure of 4C1M.

## 3. Control Method of 6-DOF Magnetic Levitation System Using Current Vector Control

The core of the control method of the 6-DOF magnetic levitation system proposed in this paper is to use the current vector control of the existing permanent magnet motor, not to apply a complicated control method for 6-DOF motion. This is an integrating control method of 6-DOF motion (levitation, rotation, tilt (Roll–Pitch), X–Y axis movement) by controlling the d–q-axis current through the current vector control of each part of the stator divided into 4 parts as shown in Figure 2.

### 3.1. Control of the Magnetic Levitation

For the levitation of the rotor, a Halbach-array permanent magnet was applied to the rotor. As shown in Figure 3, the Halbach-array permanent magnet increases the magnetic flux in the air gap through the synthesis and cancellation of the magnetic flux generated from nearby permanent magnets and can decrease the magnetic flux in the opposite direction of the air gap. The rotor is lifted by using the repulsive force generated by the interaction between the magnetic flux generated by the permanent magnet and the magnetic flux caused by the current flowing in the stator coil. The current applied to the coil can be explained by decomposing it into two directional force components ($i_{d}$, $i_{q}$) as shown in Figure 3. The D-axis current component corresponds to the magnetic force, and it can realize the force that creates the levitation force through interaction with the permanent magnet attached to the rotor. Additionally, the q-axis current component creates the rotational force of a typical motor. $F_{Z}$ in Figure 3 means repulsive force. Since the position of the rotor can be measured in real time through the encoder, the current can be controlled to match the polarity of the magnetic flux generated from the permanent magnet for all positions of the rotor, so even if the position of the rotor changes due to rotation, stable levitation is possible by generating a certain repulsive force. The total repulsive force for rotor levitation is the sum of the repulsive forces generated in each part as shown in Equation (1). (Each part means parts of the stator divided into 4 parts.)
(1)$$F_{Z}(t)=F_{ZA}(t)+F_{ZB}(t)+F_{ZC}(t)+F_{ZD}(t)$$

The control block diagram using the encoder and gap sensor placed in the magnetic levitation system for rotor levitation is shown in Figure 4. First, current is applied through the inverter to generate the levitation force of the system by calculating the error between the levitation position command through each controller corresponding to each part of the stator and the position measured from the position sensor; Equation (2) is the overall levitation command of the system, and Equation (3) is the overall levitation position measured from each gap sensor. $levi_{A,B,C,D}$* represents the levitation control command of each stator part, and $levi_{pos.\;A,B,C,D}$ represents the position sensing result of the gap sensor.
(2)$$Levitaion_{m}{}^{*}=\frac{levi_{A}{}^{*}+levi_{B}{}^{*}+levi_{C}{}^{*}+levi_{D}{}^{*}}{4}$$
(3)$$Levitaion_{m}=\frac{levi_{pos.A}+levi_{pos.B}+levi_{pos.C}+levi_{pos.D}}{4}$$

The overall levitation command ($Levitation_{m}$*) is the average value of the levitation commands of each controller, and the d-axis current command ($i_{d}$) is calculated through a current vector control algorithm so that each command satisfies the required position. Thereafter, the d-axis voltage ($V_{d})$ command is calculated by using the PI current controller, based on the current command and the actual current measured by the current sensors. This voltage command is converted into a three-phase voltage command through coordinate transformation, and a voltage is applied to the magnetic levitation system through a space vector voltage modulation method in the inverter to generate the levitation force required by the system.

### 3.2. Control of the Tilt (Roll–Pitch)

The control of the tilt (Roll–Pitch) is possible by controlling the levitation position by generating different forces on each part of the stator divided into four parts. To control the tilt (roll–pitch), measure the distance(r) of the non-contact position sensors in the sensor base and the levitation position as shown in Figure 5, and calculate the tilt of the rotor using Equation (4).
(4)$$\begin{array}{l} \theta_{Pitch}=\arctan\frac{levi_{pos.A}-levi_{pos.C}}{r}\\ \theta_{Roll}=\arctan\frac{levi_{pos.B}-levi_{pos.D}}{r} \end{array}$$

In order to control the roll–pitch by giving the difference in the levitation position of each part, the measurement values of two non-contact position sensors are needed to calculate the tilt command. The control block diagram for tilt control is shown in Figure 6. As in the previous levitation control method, when the tilt command ($Roll,Pitch_{m}$*) is input, the current command ($i_{roll,pitch}$*) for each part is output through the roll and pitch controller. Then, the current command is converted into a d-axis current command ($i_{d})$ through the current vector control algorithm, and the required voltage is applied to the inverter through the current controller.

### 3.3. Method of Levitation and Tilt Sensing Using Gap Sensors

Since the 6-DOF magnetic levitation system of this paper has no shaft and bearings, and when levitating, it can easily lose position information due to disturbance and force imbalance, leading to derailment and uncontrollable situations. Therefore, stable levitation control of the magnetic levitation system and control to maintain a balance of forces are absolutely necessary. In the case of magnetic levitation using repulsive force, the repulsive force decreases as the gap between the stator and the rotor increases. Therefore, as shown in Figure 7, the limits and commands of levitation, roll and pitch control should be calculated using the main sensing range of the gap sensor. Considering the Z-axis sensing range of the sensor in Figure 7, each control command and limit for the control of levitation and tilt (Roll Pitch) was set as shown in Figure 8, and through this, magnetic levitation and tilt control were performed.

### 3.4. Control of the Rotation

Since the 6-DOF magnetic levitation system in this paper is intended to replace the existing spin coater, it is essential to rotate at a constant speed for uniform coating of photoresist. The rotation control of the 6-DOF magnetic levitation system is the same as the rotation control of a general permanent magnet motor using current vector control, but since there is no shaft and bearing, the position of the rotor is determined using non-contact encoders located on each part of the stator. The total torque of the rotor is the sum of the torques generated in each, and the speed of the rotor used to control the rotational speed of the magnetic levitation system can be calculated as shown in Equation (5). $\omega_{m,A,B,C,D}$ represents the value for the rotational speed of each part, and the force is determined by the q-axis current $i_{q}$* component in Figure 9.
(5)$$\omega_{m}=\frac{\omega_{m,A}+\omega_{m,B}+\omega_{m,C}+\omega_{m,D}}{4}$$

The control block diagram for 6 degree of freedom rotation control is shown in Figure 9. Similar to the rotation speed control of a general motor, after measuring the position information of the rotor through the encoder, the current command is generated through the speed controller. Then, based on the q-axis current command generated through the current vector control, the voltage is applied to the inverter.

### 3.5. Control of the X–Y Axis Movement

In order to simplify the system for moving the nozzle used in the EBR process after the photoresist coating process on the semiconductor wafer, it is necessary to move and control the X–Y axis (position) of the magnetic levitation system. When the rotational speed is differently controlled for each part, a difference in centrifugal force occurs in each part of the rotor, and as a result, the rotor moves in one direction due to the imbalance of the centrifugal force [12]. That is, the X–Y axis movement control of the rotor of the magnetic levitation system proposed in this paper generates a difference in rotational force for each part of the rotor as shown in Figure 6 and Figure 10, artificially generating eccentricity in the X or Y axis so that it is to generate the force that can move the rotor. $F_{\theta ,ABDC}$ represents the rotational force generated from the stator of each part, and $F_{move,X,Y}$, which generates the force of movement, is generated by using the difference between the two forces corresponding to the X–Y axes.
(6)$$\begin{array}{l} F_{move,X}=F_{\theta ,A}-F_{\theta ,C}\\ F_{move,Y}=F_{\theta ,B}-F_{\theta ,D} \end{array}$$

### 3.6. Method of Rotation and X–Y Axis Movement Sensing Using Non-Contact Encoders

Non-contact encoders that use the method of sensing the encoder scale attached to the rotor have a limited sensing range for the X–Y axis. However, even if changes (errors) occur in the position of the encoder scale and sensor according to the movement of the X–Y axes of the rotor, position control is possible within the sensing range as shown in Figure 11. Therefore, as shown in Figure 12, an X–Y axis movement control process was designed that can control the X–Y axis movement within the sensing range of the non-contact encoder and which restores the rotor back to the sensing range when it is out of the sensing range.

### 3.7. Integrated Driving Control of 6-DEF Magnetic Levitation System

Above, the method of controlling the levitation and tilt control, rotation and X–Y movement within the sensing range of each sensor has been described. For the integrated driving control of the 6-DOF magnetic levitation system based on this, a system integrated control algorithm considering the limit of the sensing range is required. Levitation and Roll–Pitch control the force generated on each axis through the d-axis current control. To this end, the d-axis current should be generated by calculating the final command of each part of the stator divided into four parts as the sum of the tilt and magnetic levitation control commands. Rotation and X–Y axis movement control the force generated on each axis through q-axis current control. As in the previous method, the q-axis current is generated by calculating the final command of each part of the stator as the sum of the rotation and X–Y axis movement control commands. Figure 13 shows d–q axis control block diagram of each part of the stator divided into 4 parts for the integrated drive control of the 6-DOF magnetic levitation system.

## 4. Control Simulation of 6-DOF Magnetic Levitation System

### 4.1. Six Degrees of Freedom Magnetic Levitation System Modeling

For control simulation of the 6-DOF levitation system, simulation modeling consisting of a 6-DOF controller, current vector controller and 6-DOF levitation system was designed as shown in Figure 14a. The parameters of the designed magnetic levitation system are shown in Table 1. The 6-degrees-of-freedom levitation system is designed in the form of 4 motors to realize a real system consisting of a 4-segmented stator and a single rotor. The six force commands output through each controller were converted into respective current components by distributing the force commands to enable simultaneous control as shown in Figure 14b. Additionally, as shown in Figure 14c, the d–q axis current command is distributed according to each part of the system. A total of four pairs of d–q axis current commands are generated for six-degree-of-freedom motion control (levitation, rotation, tilt, translation). That is, an algorithm was implemented to generate 4 d–q axis current commands by combining 16 current commands to generate 8 current commands for generating a total of 8 forces of the 6-DOF magnetic levitation system [13]. Figure 14d shows the internal variables of the current controller composed of the vector controller in Figure 14a. The current command of $i_{dq}$ is converted into a 2–phase voltage command, and the d–q axis voltage command is converted into a 3–phase voltage command through coordinate conversion and input to the inverter through the gate to generate the output of the grid. Using the current value applied to the motor through each inverter configured as shown in Figure 14a and motor modeling, the current controller re-outputs the voltage command as much as the difference between the current command and the feedback current.

The control command entered into the 6-DOF Controller in Figure 14a converts the force generated through the 6-DOF System (Motor Mech Plant) into the position result value according to the load to calculate the output value for the control command.

### 4.2. Control Algorithm Modeling for Out-of-Control and Derailment Prevention

As explained in Section 3.3, since the 6-DOF magnetic levitation system is composed of only the magnetic force of magnets and stators without shafts and bearings, position signal information may be lost or derailment may occur during levitation. Therefore, additional modeling is required to secure stable position control performance. The position controller in Figure 14a includes a recovery controller and limiter modeling to ensure stable position control performance, as shown in Figure 15. As shown in Table 2, the limits were designed by applying the maximum sensing distance of the sensor applied to the magnetic levitation system to simulation modeling.

The system control algorithm was configured by applying an algorithm including derailment prevention and reset functions to the levitation, movement, roll and pitch controllers in the manner shown in Figure 15.

### 4.3. 6-DOF Magnetic Levitation System Simulation Result

Figure 16 shows the feedback result value for 6-DOF control command. Figure 15 shows the control simulation results for rotation speed command 50 rpm, levitation command 0.2 mm, tilting command (Roll–Pitch) 0.1 deg and X–Y axis movement command 0.05 mm using control simulation modeling.

As shown in Figure 16b, it can be confirmed that overshooting occurs because a lot of force is required for the initial levitation of the magnetic levitation system, and position control proceeds afterwards. In the case of the position control result values of (a,c–f) in Figure 16, it can be confirmed that the control is relatively stable although it contains some ripple components.

## 5. Performance Test of 6-DOF Magnetic Levitation System Using a Prototype

To verify the validity of the 6-DOF magnetic levitation system design and control strategy proposed in this paper, a performance test was conducted using a prototype, as shown in Figure 17.

Figure 18a,b are the performance test results for levitation and rotation by inputting a speed command of 50 rpm and a levitation command of 0.2 mm in the system’s stopped state. At this time, the tilt and movement control commands were set to 0. The performance test for tilt and movement was performed within the permissible range of the sensor, and when the rotor was lifted by 0.1 mm and rotated at 50 rpm, the roll–pitch and X–Y axis movement commands were applied to perform the tilt and movement performance. The test results are shown in Figure 18c–f.

In the case of the speed control in Figure 18a, a large overshoot occurs because a lot of input current is required to overcome the frictional force of the ground at the initial start. However, it can be confirmed that it converges at a relatively stable average speed.

Additionally, the result values of (a–f) in Figure 18 appear as if many ripple components occur as a whole (average 0.2 mm, 0.2 degree). However, in the case of precise control, even a small ripple component can greatly affect the result value, and it can be confirmed that the average value converges to the command relatively well.

As shown in Figure 15, the experiment as shown in Figure 18 was conducted by applying the restoration algorithm for derailment prevention, and it can be confirmed through the result graph that the continuity of the position signal is not broken. In addition, in order to secure the accuracy and reliability of precise control, the result of the experiment was derived as the data without using a filter.

## 6. Conclusions

This paper is about a 6-DOF magnetic levitation system to replace the spin coater used in the wafer process during the semiconductor production process. By proposing the design of 6-DOF magnetic levitation system applying axial flux permanent magnet motor without shaft and bearing and the 6-DOF integrated driving control method (levitation, rotation, tilt (Roll–Pitch), X–Y axis movement) using non-contact position sensors, it can solve the problems of the existing spin coater and also simplify the nozzle position control system of the EBR process. This is thought to be applicable and expandable not only to semiconductor process systems, but also to various industrial fields that require 6-DOF drive and clean production processes.

## Figures and Tables

**Figure 1 sensors-23-00905-f001:**
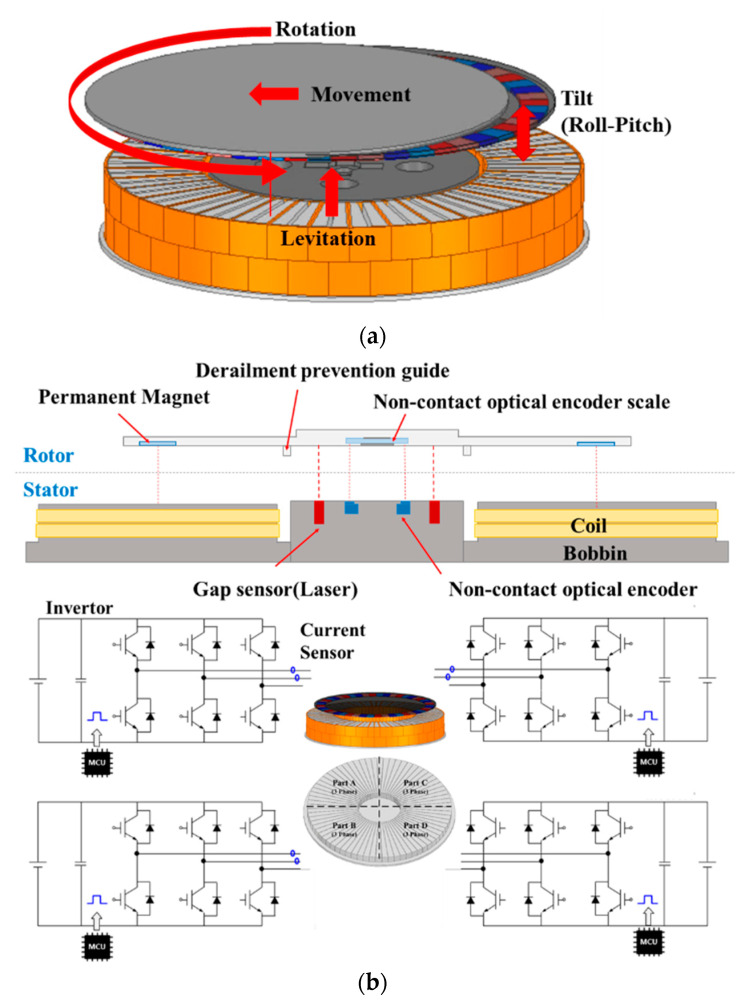
6-DOF magnetic levitation system: (**a**) Axial flux permanent magnet Motor of (**b**) 4C1M system.

**Figure 2 sensors-23-00905-f002:**
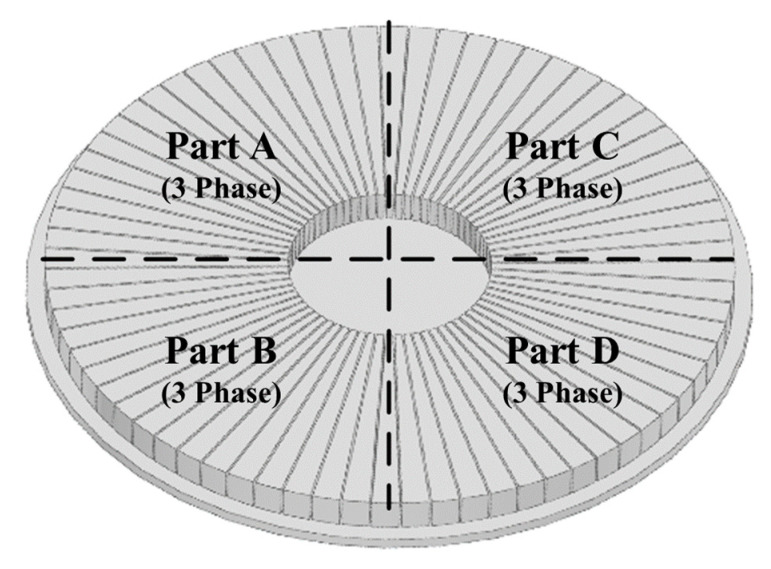
Stator divided into 4 parts (A–D).

**Figure 3 sensors-23-00905-f003:**
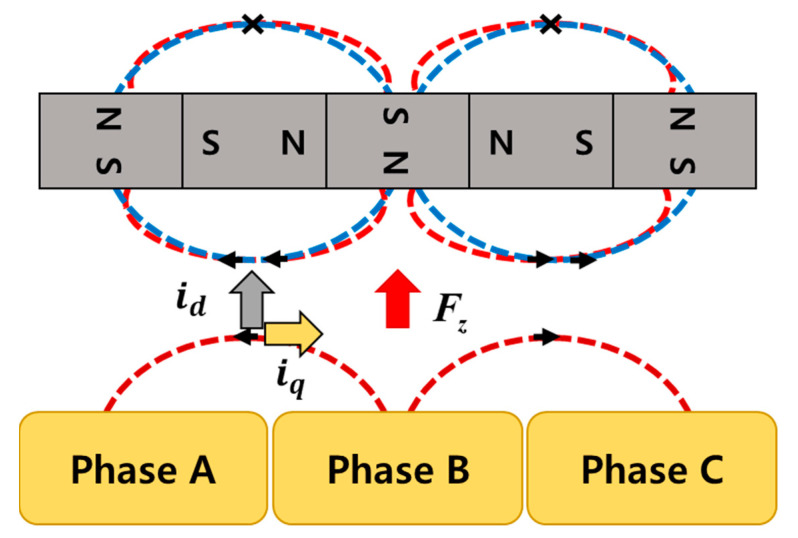
Halbach-array permanent magnet.

**Figure 4 sensors-23-00905-f004:**
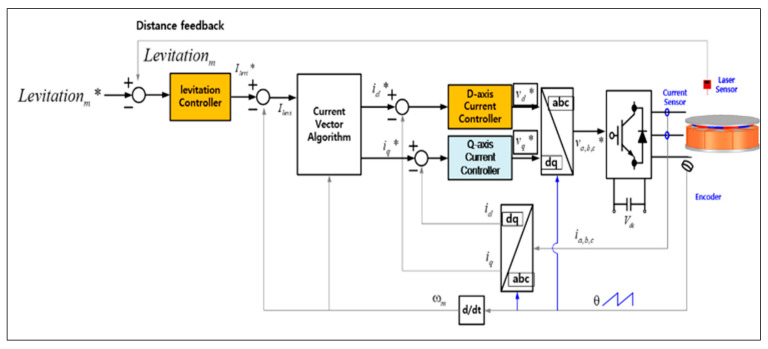
Control block diagram for the levitation (* is command value).

**Figure 5 sensors-23-00905-f005:**
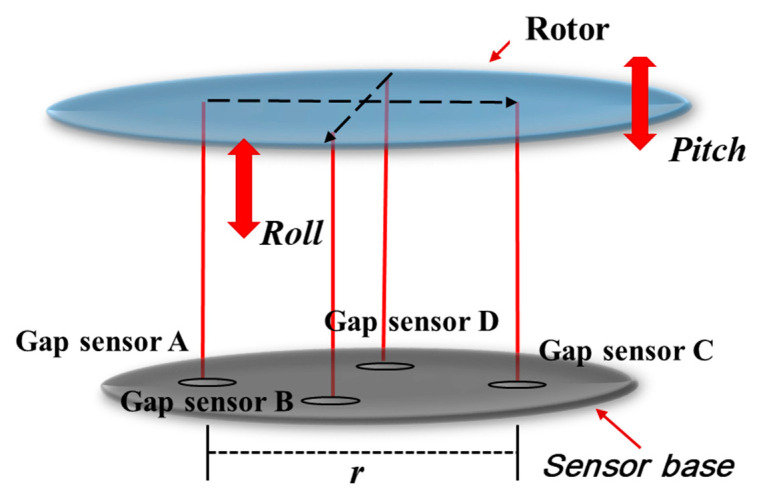
Conceptual diagram of Magnetic levitation system with non-contact position sensors.

**Figure 6 sensors-23-00905-f006:**
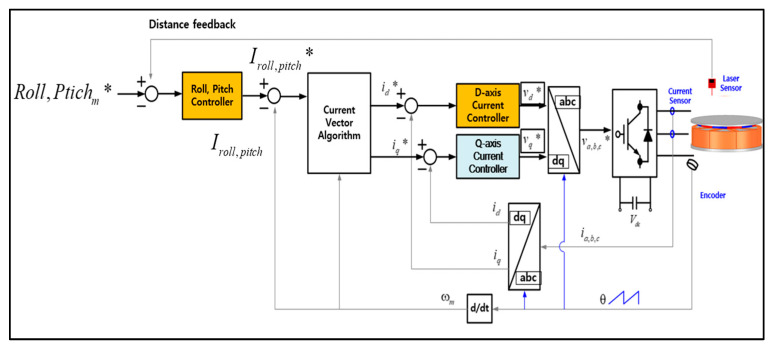
Control block diagram for the tilt (Roll–Pitch) (* is command value).

**Figure 7 sensors-23-00905-f007:**
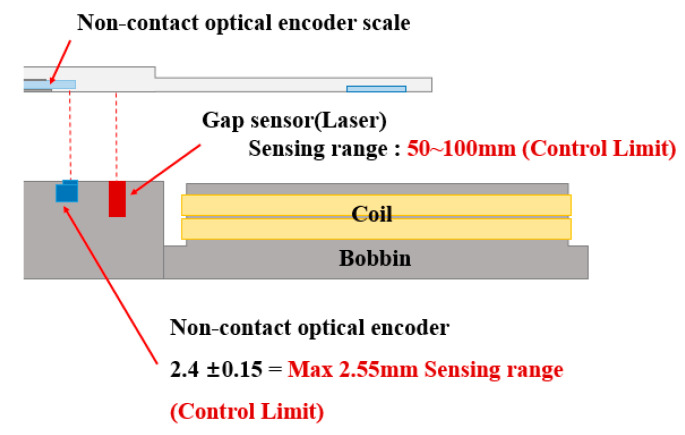
Control limit setting using Z-axis sensing range of non-contact encoder and Gap Sensor.

**Figure 8 sensors-23-00905-f008:**
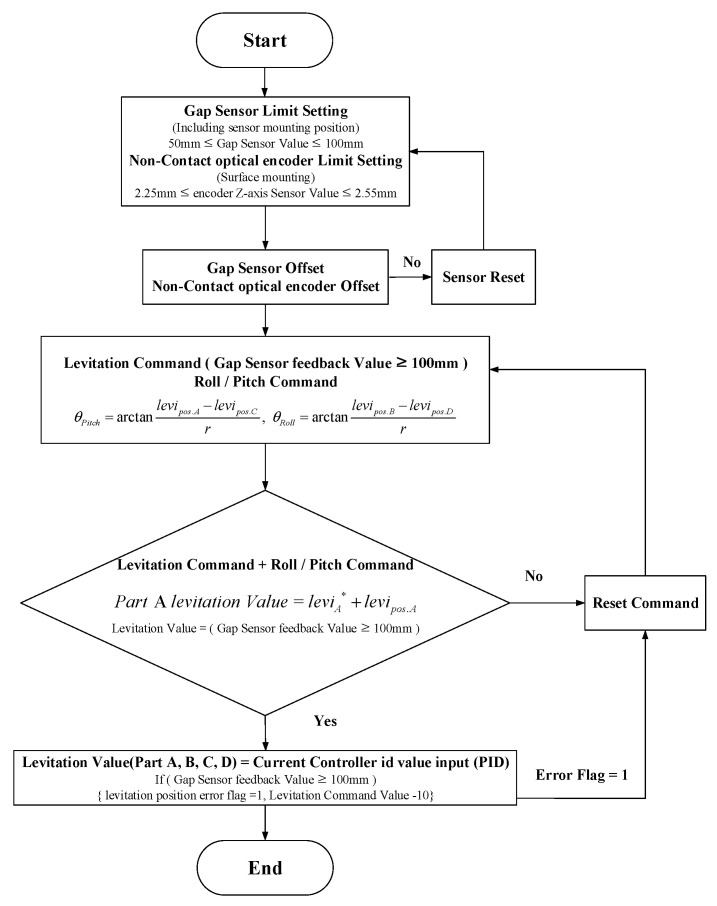
Control command calculation process for levitation and tilt (Roll–Pitch) control (* is command value).

**Figure 9 sensors-23-00905-f009:**
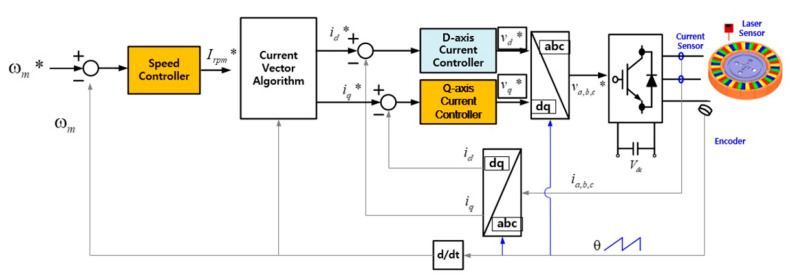
Control block diagram for the rotation (* is command value).

**Figure 10 sensors-23-00905-f010:**
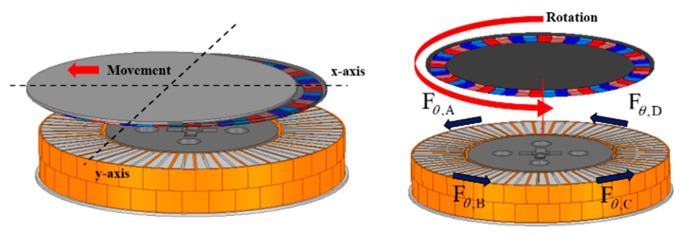
Principle of X–Y axis movement.

**Figure 11 sensors-23-00905-f011:**
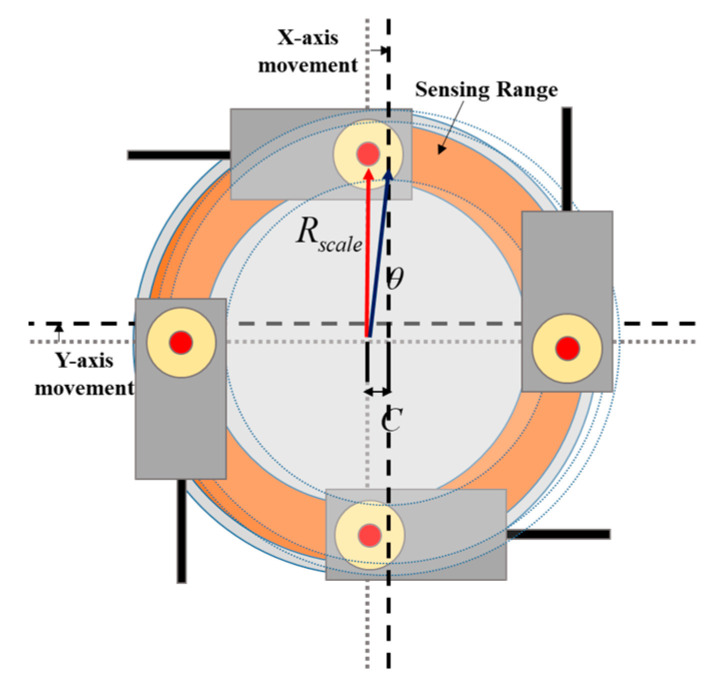
Sensing range of the non-contact encoders according to the X–Y axis movement.

**Figure 12 sensors-23-00905-f012:**
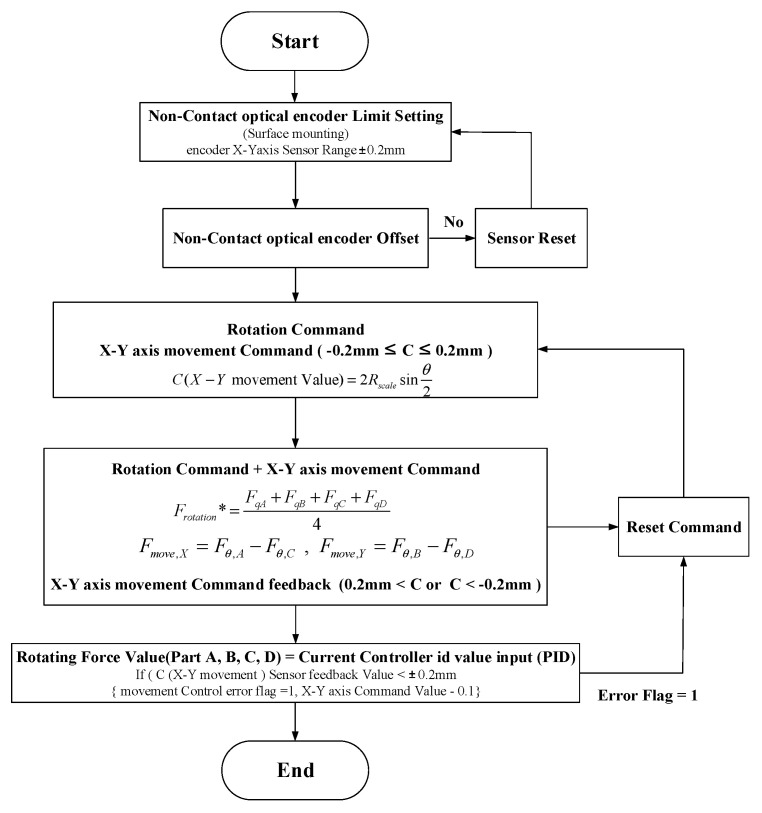
Control command calculation process for X–Y axis movement (* is command value).

**Figure 13 sensors-23-00905-f013:**
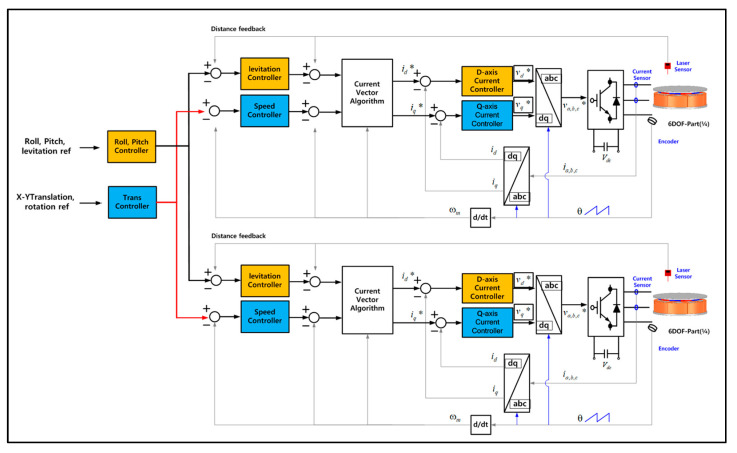
Control block diagram for the integrated driving control of the 6−DOF magnetic levitation system (* is command value).

**Figure 14 sensors-23-00905-f014:**
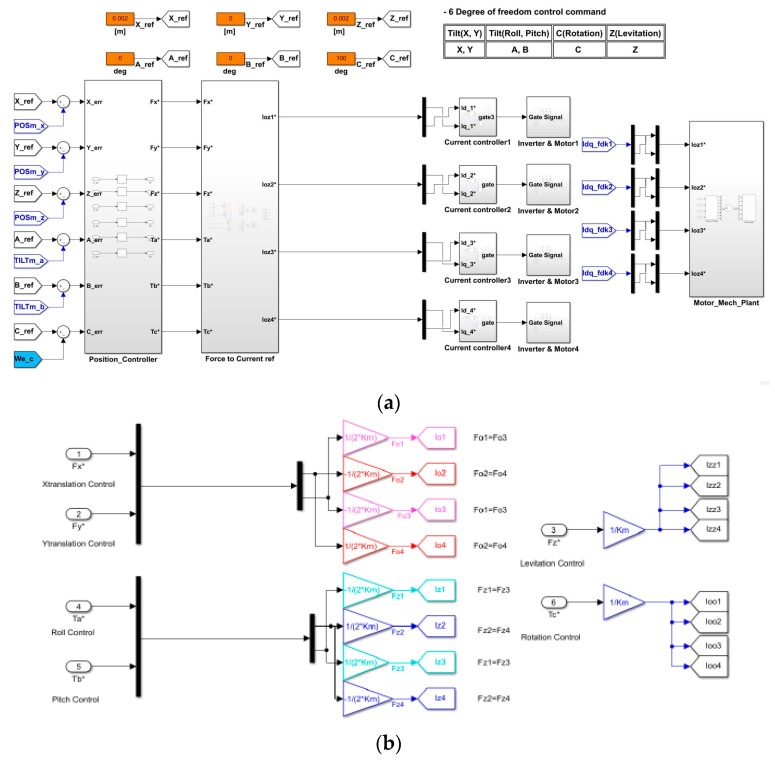
Control simulation modeling of 6-DOF levitation system: (**a**) Simulation modeling; (**b**) Current commands for each part control; (**c**) Distributing the current command of 4 parts of the magnetic (**d**) Current controller (* is command value).

**Figure 15 sensors-23-00905-f015:**
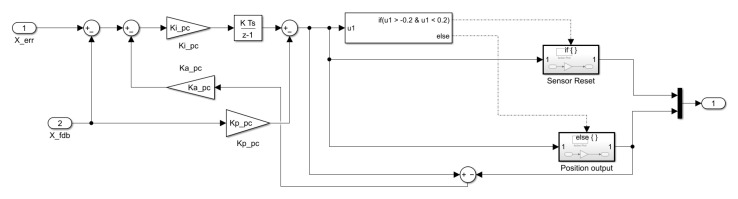
Restoration control algorithm to prevent derailment of magnetic levitation system (X-axis movement).

**Figure 16 sensors-23-00905-f016:**
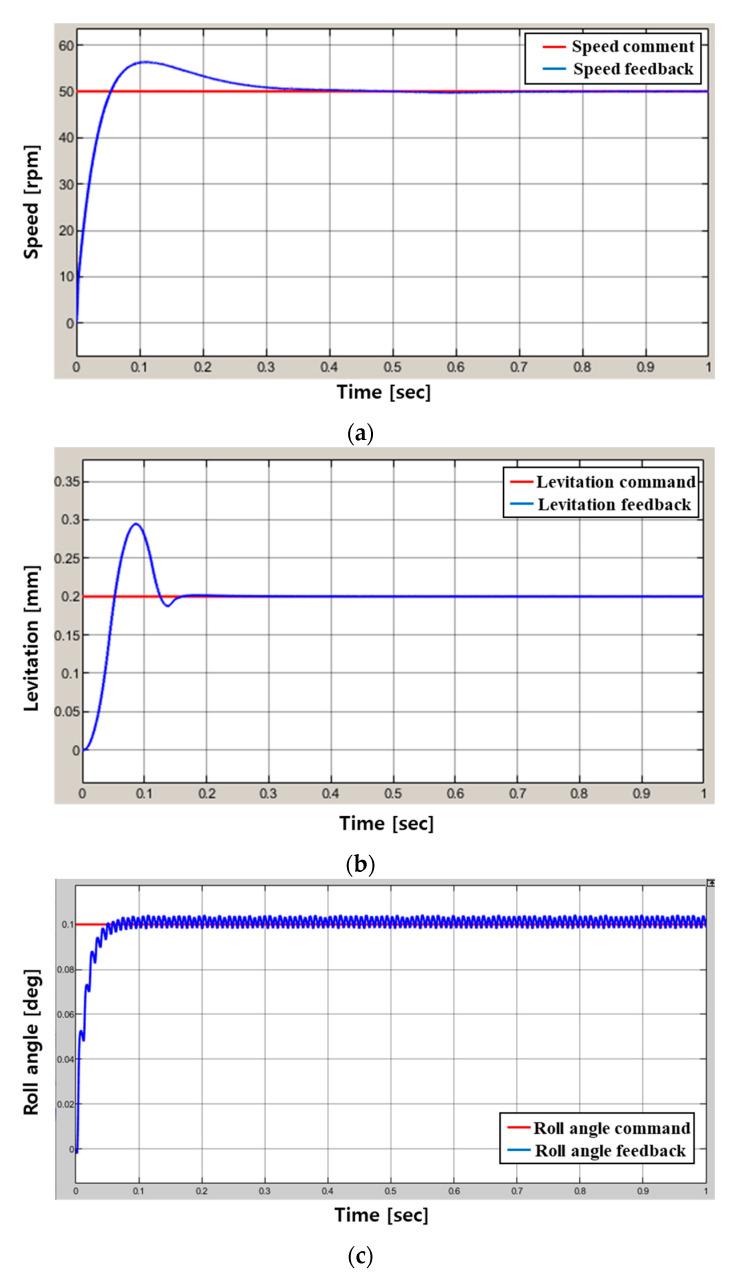
Results of the control simulation: (**a**) Rotation, (**b**) Levitation, (**c**) Roll, (**d**) pitch, (**e**) X-axis movement, (**f**) Y-axis movement.

**Figure 17 sensors-23-00905-f017:**
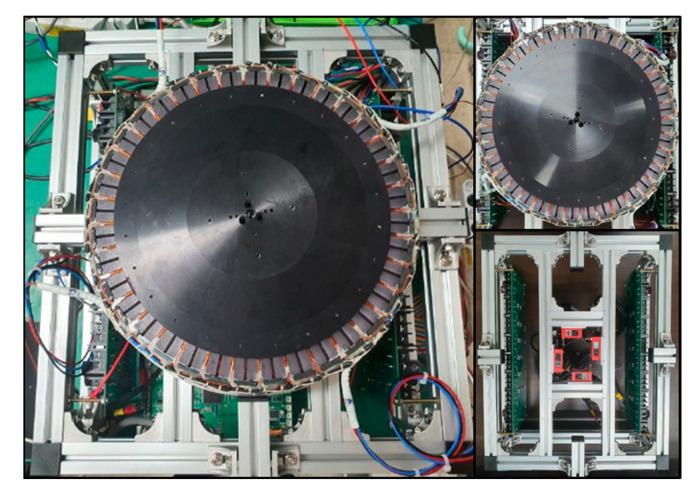
Prototype of 6-DOF magnetic levitation system.

**Figure 18 sensors-23-00905-f018:**
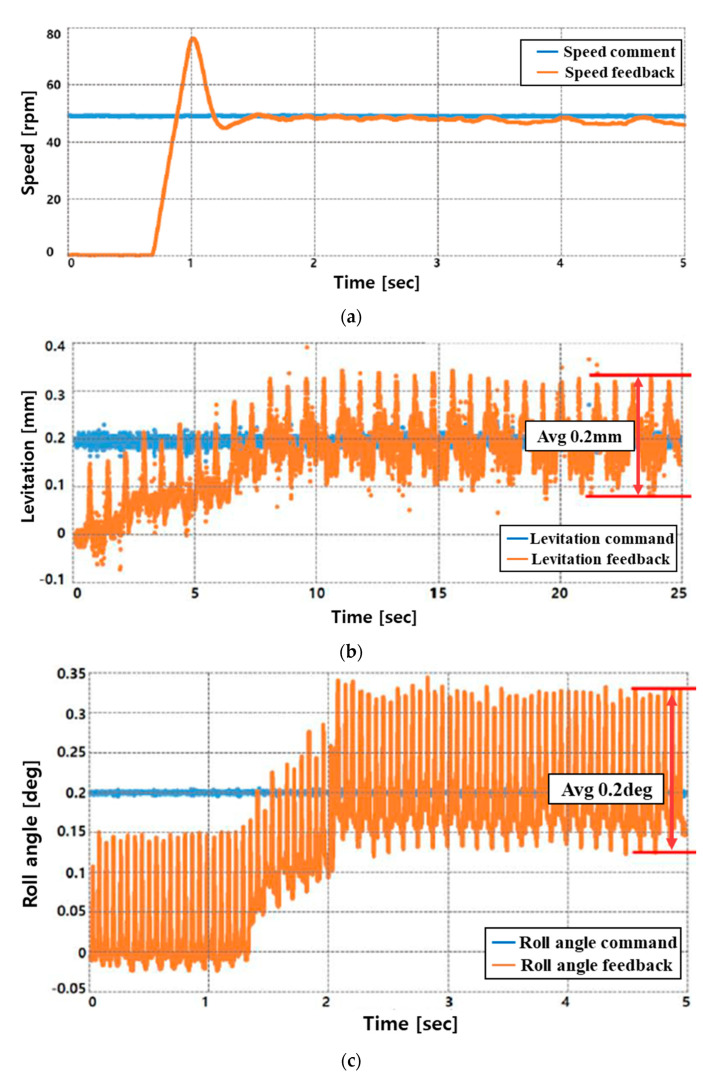
Results of the performance test using a prototype: (**a**) Rotation, (**b**) Levitation, (**c**) Roll, (**d**) pitch, (**e**) X-axis movement, (**f**) Y-axis movement.

**Table 1 sensors-23-00905-t001:** Six-degrees-of-freedom magnetic levitation system parameters.

System Classification	Variable Name	Value
Maglev Motor System Parameters	Number of System Magnet Poles	24
	Rotor Weight [kg]	1.33
	Number of Stator Coils [Turn]	15
	Phase Resistance $\left(R_{s}\right)$ [Ω] Outer Diameter of Rotor [mm]	11.5 261
	Stator Outer Diameter [mm]	316
System Controller Parameters	DC Link Voltage [V]	60
	Phase Voltage Maximum [V]	$V_{dc}/\sqrt{3}$
	Input Current Maximum [A]	10
	No Load Linkage Flux [Wb]	0.02685
	d-axis Inductance ($L_{s})$ [mH]	0.015 (1.7 (mH))
	Current Controller Period $\left(T_{cc}\right)$ [Hz]	2 kHz
	$i_{dq}$-axis P gain	$L_{s}*\left(2*\pi \right)/\left(T_{cc}\right)$
	$i_{dq}$-axis I gain	$R_{s}*\left(2*\pi \right)/\left(T_{cc}\right)$
	Carrier Frequency	0.00025

**Table 2 sensors-23-00905-t002:** Sensing limit range specification of sensor.

Sensor Classification	Variable Name	Limit Value
Non−contact encoder sensor	Encoder X–Y axis sensing range [mm]	±0.2
	Encoder Z-axis sensing range [mm]	2.25~2.55
Gap Sensor	Gap Sensing range [mm]	50~100

## Data Availability

Not applicable.

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
