# Peer review of "A Study on the Control Method of 6-DOF Magnetic Levitation System Using Non-Contact Position Sensors"

_sensors, 2023, doi:10.3390/s23020905_

Round 1

Reviewer 1 Report

This manuscript designs a six-degree-of-freedom (6-DOF) magnetic levitation system without shaft and bearings for application in semiconductor process systems to solve the problems that arise in semiconductor processes. And an integrated driving control algorithm for 6-DOF control (levitation, rotation, tilt (Roll-Pitch), X-Y axis movement) using the force of each current component derived through current vector control was proposed. The validity of the 6-DOF magnetic levitation control system was verified by simulation and experiment. It has strong practical application value and certain innovative, but there are still some details that need to be modified.

1It is recommended to give the specific meaning of all variables appearing in this manuscript.

2This manuscript has a relevant simulation analysis, is it possible to give the derivation process of the dynamics model of this magnetic levitation system?

3Are there any coupling effects between the degrees of freedom of the 6-DOF magnetic levitation system? Should it be decoupled?

4The simulation and experimental control parameter values of the 6-DOF magnetic levitation control system should be listed in a table.

5In subsection 3.3, the authors mention "Since the 6-DOF magnetic levitation system of this paper has no shaft and bearings, and when levitating, it can easily lose position information due to disturbance and force imbalance, leading to derailment and uncontrollable situations. Therefore, stable levitation control of the magnetic levitation system and control to maintain a balance of forces are absolutely necessary.“. Therefore, the simulation and experiment of anti-disturbance should be supplemented in the 4 and 5 chapters.

6In Chapters 4 and 5, there is a lack of detailed analysis of the simulation and experimental results.

7This manuscript should have a total of 6 chapters, and "5. Conclusion" should be replaced with "6. Conclusion".

Author Response

First of all, thank you for reviewing the thesis. We will respond sincerely to the contents of the review.

The entire Word file reflecting the contents of the answer was attached, and the answers to the questions were organized and recorded at the end of the document.

Reviewer 2 Report

The paper is interesting.

In my opinion abstract sentences “Recently, due to the development of semiconductor technology, high-performance memory and digital convergence technology that integrates and implements various functions into one semiconductor chip is regarded as the next-generation core technology. In the semiconductor manufacturing process, various motors are being applied for automated processes and high product reliability. However, dust and shaft loss due to mechanical friction of a general motor system composed of motor-bearing are problematic for semiconductor wafer processing.” do not fit presented in paper examinations.

Beside how without shaft transmit energy of motor– contactless? It seems to me that the sentence about the progress related to possible applications of a rotating levitating ring is relevant here (see gyroscopes). This issue should, according to expand me in the introduction instead of debatable application of the non-contact motor to technological processes. Because then it is necessary to discuss the lifting force of the motor and the automatic positioning of the motor axis after it is loaded with a technological vessel. In order to complete the research, it is then enough to provide the characteristics of the rotational speed of the levitating disk. Then, specifying the maximum rotational speed for the indicated mechanical parameters of the levitating disk.

Author Response

First of all, thank you for reviewing the thesis. We will respond sincerely to the contents of the review.

Various motors currently used in semiconductor manufacturing processes require bearings and shafts to transmit power. However, the problem of bearing and shaft wear inevitably occurs in the motor, and to solve this problem, a suction device and dust blocking structure are additionally installed and used. This paper describes a control algorithm that can design and drive a motor system using a magnetic levitation method to solve the dust problem without using such an additional structure.

In the case of the magnetic levitation system proposed in this paper, the levitation force is created through the interaction of the magnetic force generated from the magnet and the coil of the stator, and the motor system is driven using the rotational force.

The thesis was revised by inserting additional explanations on the proposed system and driving algorithm, and it is attached as an answer file.

thank you.

Round 2

Reviewer 1 Report

The quality of the paper has been greatly improved, and I agree to publish.

Author Response

Your help in improving the quality of this document is greatly appreciated.

And I will correct the awkward English spelling to improve the completeness of the paper.

Reviewer 2 Report

Please rethink the presentation of the electronic circuits shown in Figure 1 and suggest a readable version. I see three parallel end stages of a current key -- of which only one key is controlled. It is evident error of presentation as in figure 2 we can see “3-phase” label. Also, I see on fig 1 confusing labels “Invertor” and “Current Sensor” connected to current key output while in figure 4 one can guess that current sensor is non-contact type and connected to “abc” unit? “abc” unit is not described in text. What is “abc”? Figure 4 – what function is of unit “+O-“

Please show in figure where is D-axis, Q-axis mentioned in fig 4.

Thank you for figure 13 – but I can see two basis controlled is it true?

In my opinion:

figure 1 should present simplified scheme of mechanical construction,

figure 2, should present physical phenomes and interfaces between mechanical and electronic subsystems,

figure 3 that is lacking, should present a simplified scheme of electronic system and its components,

Then figures 4, 6, 9 can address specific tasks.

Author Response

Thank you very much for your review.
I will sincerely answer all questions and amend the contents.
